# Estimating the uncertain effect of the COVID pandemic on drug overdoses

**Ali Moghtaderi**[1]*, **Mark S. Zocchi**[2], **Jesse M. Pines**[3,5], **Arvind Venkat**[3,5], **Bernard Black**[4,6]

**1** Milken Institute School of Public Health, George Washington University, Washington, DC, United States of America, **2** Heller School for Social Policy and Management, Brandeis University, Waltham, Massachusetts, United States of America, **3** US Acute Care Solutions, Canton, Ohio, United States of America, **4** Pritzker School of Law, Northwestern University, Evanston, Illinois, United States of America, **5** Department of Emergency Medicine, Allegheny Health Network, Pittsburgh, Pennsylvania, United States of America, **6** Kellogg School of Management, Northwestern University, Evanston, Illinois, United States of America

* moghtaderi@gwu.edu

## Abstract

### Objective

U.S. drug-related overdose deaths and Emergency Department (ED) visits rose in 2020 and again in 2021. Many academic studies and the news media attributed this rise primarily to increased drug use resulting from the societal disruptions related to the coronavirus (COVID-19) pandemic. A competing explanation is that higher overdose deaths and ED visits may have reflected a continuation of pre-pandemic trends in synthetic-opioid deaths, which began to rise in mid-2019. We assess the evidence on whether increases in overdose deaths and ED visits are likely to be related primarily to the COVID-19 pandemic, increased synthetic-opioid use, or some of both.

### Methods

We use national data from the Centers for Disease Control and Prevention (CDC) on rolling 12-month drug-related deaths (2015–2021); CDC data on monthly ED visits (2019-September 2020) for EDs in 42 states; and ED visit data for 181 EDs in 24 states staffed by a national ED physician staffing group (January 2016-June 2022). We study drug overdose deaths per 100,000 persons during the pandemic period, and ED visits for drug overdoses, in both cases compared to predicted levels based on pre-pandemic trends.

### Results

**Mortality**. National overdose mortality increased from 21/100,000 in 2019 to 26/100,000 in 2020 and 30/100,000 in 2021. The rise in mortality began in mid-to-late half of 2019, and the 2020 increase is well-predicted by models that extrapolate pre-pandemic trends for rolling 12-month mortality to the pandemic period. Placebo analyses (which assume the pandemic started earlier or later than March 2020) do not provide evidence for a change in trend in or soon after March 2020. State-level analyses of actual mortality, relative to mortality predicted based on pre-pandemic trends, show no consistent pattern. The state-level results

**Data Availability Statement:** All the data on overdose deaths were uploaded as part of the resubmission package. The national CDC non-fatal overdoses database was also uploaded. The

emergency department visit from the national staffing group belongs to the U.S. Acute Care Solutions and will remain confidential. The data use agreement in place. Individuals can reach Pablo Colden (celedonp@usacs.com) in U.S. Acute Care Solutions for reasonable data requests.

**Funding:** The authors received no specific funding for this work.

**Competing interests:** The authors have declared that no competing interests exist.

support state heterogeneity in overdose mortality trends, and do not support the pandemic being a major driver of overdose mortality.

**ED visits**. ED overdose visits rose during our sample period, reflecting a worsening opioid epidemic, but rose at similar rates during the pre-pandemic and pandemic periods.

## Conclusion

The reasons for rising overdose mortality in 2020 and 2021 cannot be definitely determined. We lack a control group and thus cannot assess causation. However, the observed increases can be largely explained by a continuation of pre-pandemic trends toward rising synthetic-opioid deaths, principally fentanyl, that began in mid-to-late 2019. We do not find evidence supporting the pandemic as a major driver of rising mortality. Policymakers need to directly address the synthetic opioid epidemic, and not expect a respite as the pandemic recedes.

## Introduction

U.S. drug-related overdose deaths rose sharply from 71,000 in 2019 to 93,000 in 2020, and to 107,000 in 2021 [1]. Multiple studies have linked the onset of the COVID-19 pandemic in March 2020 to rising overdose deaths [2–7], and rising drug-related emergency department (ED) visits [8–12]. While some of these studies cite greater access to synthetic opioids as a secondary contributor [5, 6], they ascribe rising mortality and drug-related ED visits primarily to the pandemic and its related lockdowns, job loss, and social isolation [13]. News stories presented a similar narrative [14–16]. However, two Illinois studies report evidence for rising opioid-related deaths beginning in late 2019, suggesting an important role for non-pandemic factors [17, 18]. In addition, we previously found no increase in ED visits across 18 U.S. states for substance abuse during the early pandemic period from March-July, 2020 [19].

We revisit the evidence on the reasons for rising overdose mortality during the pandemic period by studying a longer time period, explicitly modeling the counterfactual of what levels of overdose mortality and ED visits would have been expected based on pre-pandemic trends, and examining both national and state trends. We examine two competing hypotheses. The first is that the pandemic may have been a principal driver of higher overdose deaths. This hypothesis predicts a rise in both overdose deaths and overdose-related ED visits, beginning in March 2020 (pandemic onset). Second, rising mortality might primarily reflect greater availability of synthetic opioids. This hypothesis would predict higher mortality, and related rise in ED visits beginning prior to the onset of the pandemic.

We assess the evidence for these hypotheses, recognizing that both could be partly true. Neither hypothesis can be proven or disproven from the available data. We lack a control group that was unaffected by the pandemic, and thus do not have a causal research design. However, we draw together several strands of evidence which, taken together, support continuation of pre-pandemic trends as likely to have been a principal driver of rising overdose mortality. Conversely, we do not find evidence that the pandemic was a principal driver.

## Background

The opioid epidemic refers to overdose deaths involving opioids, including prescription and illicit opioids, which has grown rapidly since the 1990s. The opioid epidemic includes three distinct waves [20]. The first wave began in the 1990s, with most overdose deaths due to

prescription opioids. The second wave started around 2010, with heroin as the main driver. The third wave began in 2013 with rapid increases in overdose deaths involving synthetic opioids, principally fentanyl [21, 22]. High potency, lower-price fentanyl has been increasingly mixed with higher price heroin, and sometimes with other drugs. Use of fentanyl dramatically increases the risk of respiratory depression and death [23].

The onset of the third wave varied across states, with rising synthetic-opioid deaths initially concentrated in states east of the Mississippi River [24, 25]. However, synthetic-opioid deaths rose in other states beginning in 2019 [26]. The evolving nature of the forces that drive the opioid epidemic presents challenges in assessing the effect of the COVID-19 pandemic on overdose deaths.

### Evidence on overdose mortality

Many studies have reported an association between COVID-19 pandemic and overdose deaths and have suggested that the pandemic was a principal driver of rising overdose deaths [2–7]. Yet, these studies have major limitations. None has a causal research design (nor do we). Only one study uses pre-pandemic data to predict counterfactual outcomes [4]. Moreover, most studies focus on limited geographic areas and/or short time periods [2, 3, 5, 7, 17, 18].

In short, none of the prior studies provides convincing evidence that the pandemic, as opposed to continuation of pre-pandemic trends, was a primary cause of the 2020 rise in overdose deaths. Four (all but Faust et al., 2021) [4] report overdose deaths for February 2020 versus February 2019; of these three (all but t Currie et al., 2021) [5] find a rise in overdose deaths in February 2020, prior to the pandemic. Currie et al. (2021) [5] report evidence for a time-limited spike in fentanyl deaths early in the pandemic, but only for a single state. See S1 Appendix for more detailed discussion of these studies.

### Trends in overdose-related ED visits

If rising deaths reflect increased drug use, one would expect drug-related ED visits (mostly for nonfatal overdoses) to also rise. Several studies report rising opioid-related ED visits [8–12], and ascribe the rise principally to the pandemic. However, these studies have limitations similar to the fatal overdose studies. First, none model the counterfactual of what ED visit rates that would be expected during the pandemic period based on pre-pandemic trends. Second, some measure overdose-related ED visits, in part, relative to total ED visits at the same point in time [8–11]. This ratio measure is confounded by pandemic-related trends in ED visits for other causes, which fell sharply early in the pandemic and then slowly and partially rebounded [27]. Some studies also cover limited geographic areas, short time periods, or both [10–12].

Our own prior study [19] covers January 2019 through July 2020, compares pandemic to pre-pandemic overdose visit rates, and finds generally higher visit rates in 2020, but similar 2020/2019 visit ratios during the pandemic period versus earlier in 2020.

Taking these studies together, prior evidence on overdose-related ED visit rates supports rising visit rates during both the pre-pandemic and early pandemic periods but does not support a pandemic-related change in visit trends.

### Methods

We conducted a retrospective study of Centers for Disease Control and Prevention (CDC) data on rolling 12-month overdose mortality from January 2015 to December 2021, CDC monthly data on non-fatal overdose ED visits from January 2019 to September 2020 (the only period with available data), and monthly data from a national emergency physician staffing

group on overdose-related ED visits from January 2016 through June 2022. The Allegheny Health Network Institutional Review Board approved the study.

## Data sources

Data on overdose deaths comes from the CDC's National Vital Statistics System (NVSS) [1]. Provisional trailing rolling 12-month counts are reported monthly for each state and the District of Columbia *but* are initially incomplete. The preliminary data we rely on is released with an approximate 6-month lag to allow for more complete reporting. Final data are not usable for this study because this data is released with a substantial lag. However, changes from preliminary to final data are modest and should have minimal effects on the time trends we study.

National data on drug-related mortality, including how many deaths involve synthetic opioids, is available from 2015 on. State data are also available from 2015 on, but a reliable breakdown of mortality into synthetic-opioid-related versus other deaths is only available beginning in 2017. Fentanyl is included within synthetic opioids and is understood to cause most synthetic opioid deaths [3, 5, 28].

For national ED visits, we used monthly data on non-fatal, overdose-related visits from 2019 through September 2020 from EDs in the 42 states that participate in the CDC's Drug Overdose Surveillance and Epidemiology (DOSE) system during this period [29]. Overdose visits are identified based on chief complaint and primary ICD-10 discharge diagnosis. Which EDs provide data can change over time, but we have no reason to expect these changes to bias our results.

We also studied non-fatal overdose-related ED visits (measured using the primary discharge diagnosis) using monthly data from 181 EDs in 24 states staffed by a national emergency physician staffing group. This database includes fewer EDs but is available at the individual ED level and covers a longer period, from January 2016 to June 2022. We report results for all EDs in the text and results for a balanced panel of 37 EDs, are in the S1 Appendix.

## Outcomes

The main outcome for drug overdose mortality was the drug-related deaths per 100,000 population. We studied total, synthetic-opioid-related, and other drug-related deaths.

For ED visits for non-fatal drug-related overdoses, the main outcome was the ratio of pandemic-period visits to visits for the same calendar month a year earlier (for 2021 and 2022 we compare to 2019 rather than 2020 and 2021 respectively). Unlike some prior studies [8, 9], we did not study the ratio of overdose-related to all ED visits in the same month during the pandemic because the pandemic led to sharp swings in all-cause ED visits, especially for lower-acuity conditions [19, 30, 31]. For EDs staffed by the national emergency medicine group, we computed this ratio at the ED level and then averaged the ratios across EDs.

## Statistical analysis

**Overdose deaths.** We report time trends for synthetic-opioid-related, other, and total drug-overdose deaths over 2015–2021. For the first pandemic year (through March 2021), we compare actual to predicted rolling-average deaths, at both the national and state levels, using a fourth-order polynomial model, which is adequate to fit the pre-pandemic data. We treated the pandemic as starting in March 2020, because the national emergency declaration and related lockdowns began in mid-March. These policies led to greater social isolation and could have led to higher substance use, with isolation also reducing access to prompt treatment when an overdose occurs. However, our results are not sensitive to whether we treat the

pandemic effect on overdoses as beginning in March 2020 or somewhat later. We examined different polynomial orders and different starting pre-pandemic periods. We selected a fourth-order polynomial as the lowest-order polynomial that provided a reasonable fit for all overdose deaths and opioid-related deaths during the pre-pandemic period. A higher-order polynomial would only slightly improve the pre-pandemic fit but would increase the risk of overfitting and thus of generating implausible projections for the pandemic period. We report counterfactual estimates using pre-pandemic period starting either in 2015 or in 2017.

To assess whether there was a change in trend in the actual pandemic-onset month (March 2020), we compared actual-minus-predicted deaths assuming the pandemic began instead in a placebo month, using placebo months both before and after March 2020. If the pandemic were a principal cause for the 2020 rise, we would expect actual-minus-predicted deaths to be positive for March 2020 pandemic onset and to be higher for March 2020 pandemic onset compared with nearby placebo months.

We compared actual to predicted mortality at both the national and state levels. To illustrate heterogeneity across states, we report results for the 48 states with actual minus predicted mortality rates between -20 and +20 per 100,000 persons (Fig 3). Including the remaining would require a more compressed y-axis scale and would not qualitatively change the principal results from this analysis.

*Drug overdose-related ED visits*. For the national ED visit data, we compared ED visits in 2020 to visits in the same month in 2019 and computed the 2020/2019 ratio of ED visits for non-fatal drug overdoses (separately, for all overdoses and opioid-related overdoses). We also compared pre-pandemic ratios in January-February 2020, to post pandemic ratios from March-September 2020. This approach control for seasonal trends in overdose-related ED visits, which typically peak in the summer.

For the ED visit data from the national staffing group, we computed, within each ED, the ratio of monthly visits in the pandemic period (March 2020-June 2022) to visits in the same month in 2019. For pre-pandemic visits (through February 2020) we computed the ratio of monthly visits to the same month a year prior. We then averaged these ratios across EDs, weighted by ED size (based on 2019 visit volume). We graphed these monthly ratios for drug-related, opioid-related, and all other overdose ED visits, including 95% confidence intervals (CIs), using ED-size weights and standard errors clustered on ED.

## Results

### Drug overdose deaths

Table 1 provides summary statistics for all overdose deaths, and for those attributable to synthetic opioids versus other overdose deaths. Synthetic opioid deaths rose steadily over 2015–2019, trends in all overdose deaths were partially, offset by falling mortality due to other drugs.

Table 1 also includes predicted values for the first pandemic year (through March 2021) for all overdoses and synthetic-opioid overdoses, using the fourth-order polynomial model based on data starting in 2015 (2015 model) or in 2017 (2017 model). With both starting points, the model predicts a rise in synthetic-opioid deaths during the pandemic period. For 2020 as a whole, the 2015 model predicts a rate of 26.3 deaths per 100,000 population, and the 2017 model predicts a rate of 24.7, versus the observed rate of 25.5.

Fig 1, Panel A, presents 12-month rolling average mortality rates for all drug-related overdose deaths, synthetic-opioid-related deaths, and other overdose deaths, from 2015–2021. The rise during the pandemic period continues a trend that accelerated in mid 2019 toward rising synthetic opioid deaths. Death from other drug overdoses were basically flat from mid-2019 on.

**Table 1. Summary statistics: National overdose mortality.** Actual and predicted rolling 12-month drug-overdose deaths per 100,000 persons over 2015-December 2021, divided into those related to synthetic opioids and other, and predicted overdose deaths in 2020 and through March 2021 for all overdoses and synthetic opioids overdoses using fourth order polynomial fitted to pre-pandemic data starting from either 2015 (2015 model) or 2017 (2017 model) through Feb. 2020.

| Year | All overdoses (actual) | All overdoses (predicted) | Synthetic opioid (Actual) | Synthetic Opioids (Predicted) | Other |
|---|---|---|---|---|---|
| 2015 | 15.82 [15.00, 22.52] | — | 2.45 [1.83,3.05] | | 13.37 [13.14, 13.58] |
| 2016 | 18.23 [16.60,20.10] | — | 4.49 [3.13,6.15] | | 13.74 [13.46, 13.94] |
| 2017 | 21.47 [20.48,22.04] | — | 7.89 [6.56,8.96] | | 13.57 [13.06, 13.91] |
| 2018 | 21.33 [21.02,21.72] | — | 9.40 [8.93,9.80] | | 11.93 [11.21, 12.79] |
| 2019 | 21.16 [20.87,21.97] | — | 10.33 [9.78,11.35] | | 10.82 [10.62, 11.10] |
| 2020 | 25.52 [22.20,28.42] | 2015 Model: 26.27 [22.26, 32.22] 2017 Model: 24.74 [22.22, 28.09] | 14.75 [11.61,17.55] | 2015 Model: 14.17 [11.52, 17.94] 2017 Model: 14.53 [11.61, 18.65] | 10.77 [10.60, 10.90] |
| 2021 (thru March) | 29.38 [28.80,30.04] | 2015 Model: 35.71 [33.86, 37.62] 2017 Model: 29.77 [28.90, 30.67] | 18.50 [17.97, 19.12] | 2015 Model: 20.07 [18.94, 21.24] 2017 Model: 20.97 [19.74, 22.22] | 10.87 [10.83, 10.91] |
| 2021 (July thru December) | 31.55 [30.51,32.83] | — | 20.51 [19.55, 21.75] | | 11.04 [10.94, 11.12] |

Fig 1, Panel B compares actual to modeled overdose death rates. It presents three lines: actual total deaths, predicted deaths using the 2015 model, and predicted deaths using the 2017 model. The prediction line from the 2015 model was similar to the actual values in the first six pandemic months, but then became increasingly above the actual values. The line from the 2017 model was somewhat below actual values in 2020, but caught up to actual deaths in early 2021. The 2017 model predicts 97% of actual 2020 deaths and 101% of 2021 deaths (through March).

Fig 1, Panel C presents the actual and predicted mortality rates caused by synthetic opioids from both the 2015 and 2017 models. The prediction lines from the 2015 and 2017 models are similar to each other. Modeled deaths are somewhat below the actual values in the first six pandemic months, but catch up by the end of 2020 and exceed actual values in early 2021. The 2017 model predicts 98% of actual 2020 synthetic-opioid deaths, and 113% of 2021 deaths (through March).

The increasing divergence of predicted from actual overdose deaths in 2021, for both total and synthetic-opioid deaths, is consistent with the difficulty of predicting the future course of the opioid epidemic over an extended period of time.

In S1 Appendix, Fig 1, we report results from a placebo analysis, in which we assume the pandemic began earlier or later than March 2020, and report actual-minus-predicted overdose mortality rates, by placebo-onset month, over the placebo pandemic period (from the placebo-onset month through March 2021). Vertical bars indicate 95% CIs. The left-hand figures show results for all drug overdoses; the right-hand figures show results for synthetic-opioid-related overdoses. Panel A uses the 2015 model, while Panel B uses the 2017 model. Each point on the graph represents the average difference between actual and predicted values of overdose deaths

Panel A

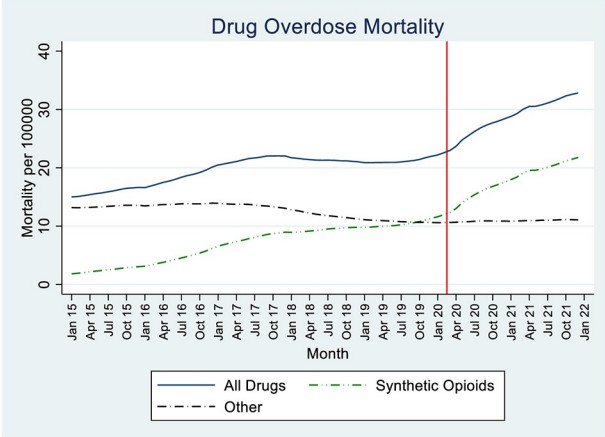

Panel B

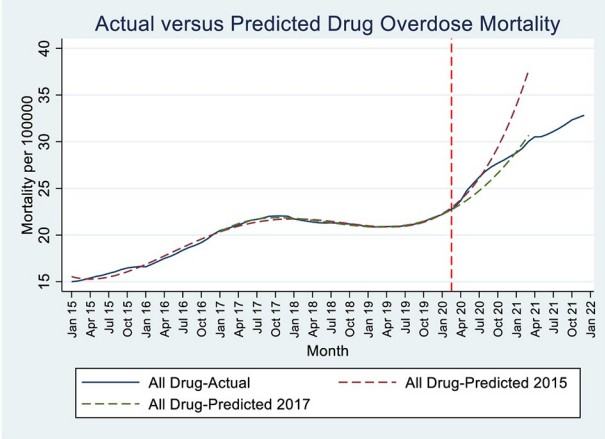

Panel C

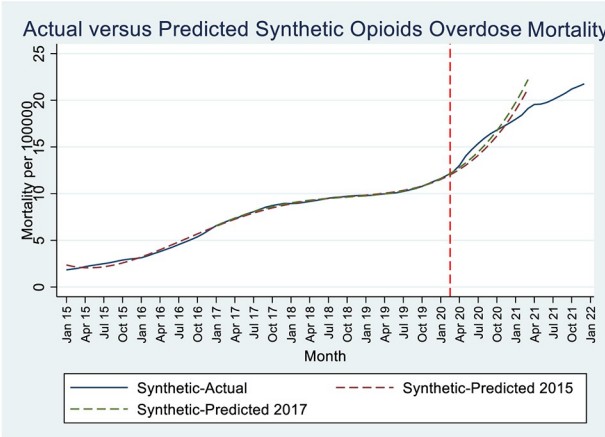

**Fig 1. Drug overdose-related deaths over 2015–2021. Panel A**. Rolling 12-month totals for national drug overdose deaths (total, synthetic-opioid related, and other) per 100,000 persons between January 2015 and December 2021. **Panel B**. Actual versus predicted overdose deaths per 100,000 persons for the first pandemic year (through March 2021), with two prediction lines based on 4th order polynomials, fitted to monthly pre-pandemic data through February 2020. One prediction line is fitted using data starting in 2015 (2015 model), other line uses data starting in

2017 (2017 model). **Panel C**. Actual versus predicted synthetic opioids deaths, with prediction lines based on the 2015 and 2017 models. **All panels**. Vertical dashed line indicates March 15, 2020, as the approximate onset of pandemic-related lockdowns. **Panel A: Total, Synthetic-Opioid Related, and Other Drug Overdose Deaths. Panel B: Actual and Predicted Drug Overdose Mortality. Panel C: Actual and Predicted Synthetic Opioids Mortality**.

during the placebo-pandemic period, with prediction based on data from the pre-placebo-pandemic period. With both models and both outcomes, the actual-minus predicted values for all overdose deaths do not change around March 2020, and generally become negative after the pandemic onset (opposite from predicted). These patterns do not provide evidence supporting a large role for the pandemic in explaining mortality trends.

Fig 2 shows the state-by-state analysis. There was substantial scatter, with some states having actual mortality close to predicted levels; some having significantly higher mortality than predicted, and others having significantly lower mortality.

We did not find any simple patterns with regard to which states showed higher-than-predicted mortality, based on either geographic region or pre-pandemic overdose rates. The scatter in the state-level results confirms the difficulty in predicting overdose deaths based on past trends, and highlights the danger in studying the role of the pandemic in overdose deaths using data from one or a few states. The overall picture, with roughly as many positive as

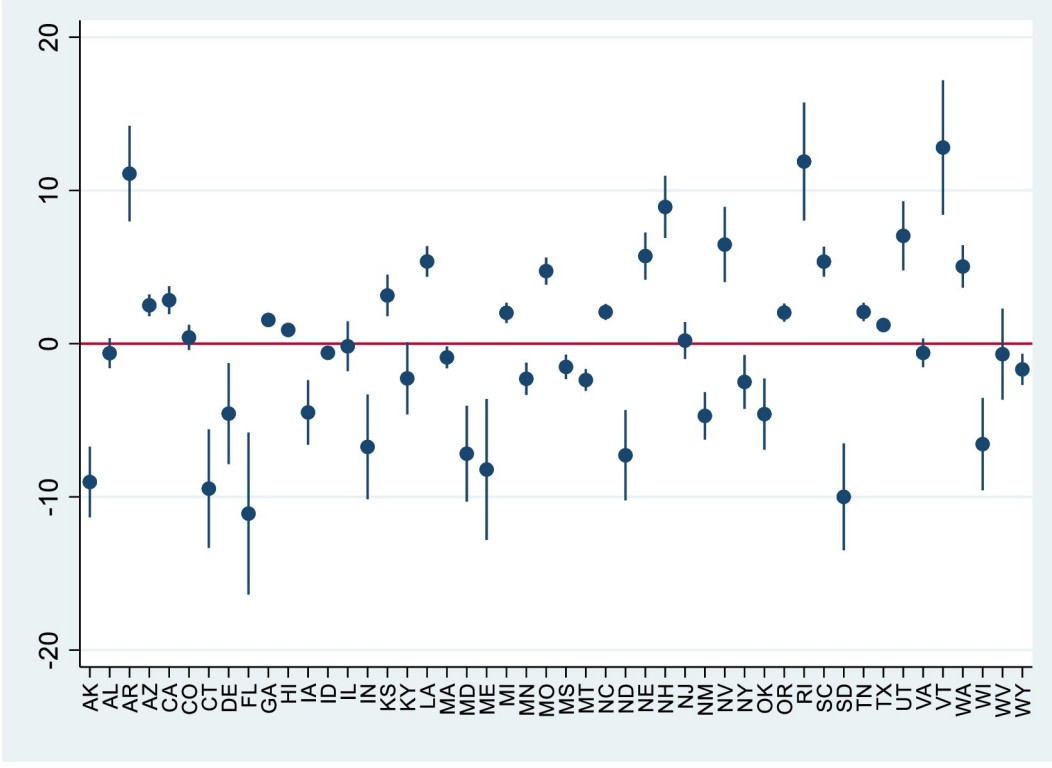

**Fig 2. State-level estimates of actual-minus-predicted drug overdose mortality per 100,000 persons during pandemic period. Note**: Outcome variable is actual-minus-predicted rolling 12-month drug overdose deaths per 100,000, using same model as in Fig 1, applied to state-level data from January 2015 through March 2021. Small vertical bars show 95% confidence intervals (CIs). 3 outlier states with |actual–predicted mortality| > 20 per 100,000 people omitted from graph. Omitted states are: District of Columbia, Ohio, and Pennsylvania.

negative point estimates, does not support a large role for the pandemic in driving state-level overdose deaths.

In sum, at both the state and the national level, the evidence supports a large role for continuation of pre-pandemic trends in explaining rising overdose deaths during 2020 and 2021, and conversely does not provide evidence that the pandemic played a major role.

## ED visits

The left-hand columns of Table 2, provide summary statistics for national 2020/2019 ratios for ED visits for all causes and for all drug overdoses, divided into opioid-related and other drug overdoses, for the period with available data (January—September 2020). All-cause ED visits were higher in the pre-pandemic months of 2020 (January-February) than in the corresponding months in 2019 (ratio: 1.16) but dropped during the pandemic period (ratio: 0.80). All drug overdose visits, especially opioid-related overdose visits, were substantially higher in the pre-pandemic months and dropped somewhat in the early pandemic period but less sharply than visits for all causes. The 2020/2019 ratios for all drug overdose visits, opioid-related visits, and other overdose visits were higher in the pre-pandemic months (1.31, 1.46, and 1.16 respectively) than during the pandemic month (1.11, 1.28, and 0.79).

The right-hand columns of Table 2 report year-over-year ratios for overdose-related ED visits from the national emergency physician staffing group for the pre-pandemic period from January 2017 thorough February 2020, and the pandemic period from March 2020 through June 2022. All cause ED visit ratios were slightly higher during the pre-pandemic period, versus a year earlier (ratio: 1.02), dropped sharply early in the pandemic, and then partly recovered. The ratios for all drug overdoses and opioid-related overdoses were above 1.00 during the pre-pandemic period (1.10 and 1.11 respectively). All drug overdose ED visits dropped somewhat early in the pandemic, but then were somewhat comparable to pre-pandemic ratios after 2021. Opioid overdose visit ratios were somewhat higher during the balance of 2020 than the average ratio over January 2017-February 2020; but showed little change relative to the immediate pre-pandemic months (January-February 2020; see Fig 3, Panel B). ED visits for all other overdoses dropped in the early pandemic months, but recovered, and by 2021 exceeded pre-pandemic levels. The year-over-year ratios, which are consistently above 1.00, but on the

**Table 2. Drug-overdose related ED visits: Ratios of pandemic to pre-pandemic rates.** *Left hand columns*: CDC data from 42 states for ratio of non-fatal drug-related ED visits in Jan.-Sept. 2020. Table shows ratio of visits during 2020 to visits during same month in 2019. *Right hand columns*: Data from ED staffing company on drug-related ED visits to 181 EDs in 24 states over January 2017-June 2022. Table shows average for indicated periods of monthly ratios of visits in January 2017-February 2020 to visits in the same months a year earlier and March-December 2020 and January 2021-June 2022 to visits in the same months in 2019. Ratios are computed at the ED level and then averaged across EDs within each month. 95% confidence intervals are reported in brackets. Last row shows average ratio of opioid/related ED visits to all overdose visits for the months in the indicated period.

| | National Data (CDC) | | ED staffing company data | | | |
|---|---|---|---|---|---|---|
| Period | Jan-Feb, 2020 | Mar.-Sept., 2020 | Jan 2017- Feb 2020 | Mar-Dec 2020 | Jan 2021-June 2022 | Jan 2022-June 2022 |
| Base Period | Same month, year earlier | | Same month, year earlier | Same month in 2019 | | |
| All Visits | 1.16 | 0.80 | 1.02 [1.01, 1.03] | 0.76 [0.75, 0.78] | 0.89 [0.87,0.90] | 0.87 [0.85,0.90] |
| All Drug Overdose Visits | 1.31 | 1.11 | 1.10 [1.08, 1.13] | 1.07 [1.01, 1.12] | 1.16 [1.10, 1.23] | 1.18 [1.11, 1.25] |
| Opioid-related Visits | 1.46 | 1.28 | 1.11 [1.07, 1.15] | 1.22 [1.11, 1.33] | 1.18 [1.07, 1.28] | 1.06 [0.95,1.16] |
| Other Drug Visits | 1.16 | 0.79 | 1.12 [1.10, 1.14] | 1.03 [0.98, 1.08] | 1.19 [1.13, 1.25] | 1.26 [1.18,1.34] |
| Opioid-related/all drug overdose visits | 30.5% | 34.1% | 29% | 29% | 25% | 21% |

**Panel A**

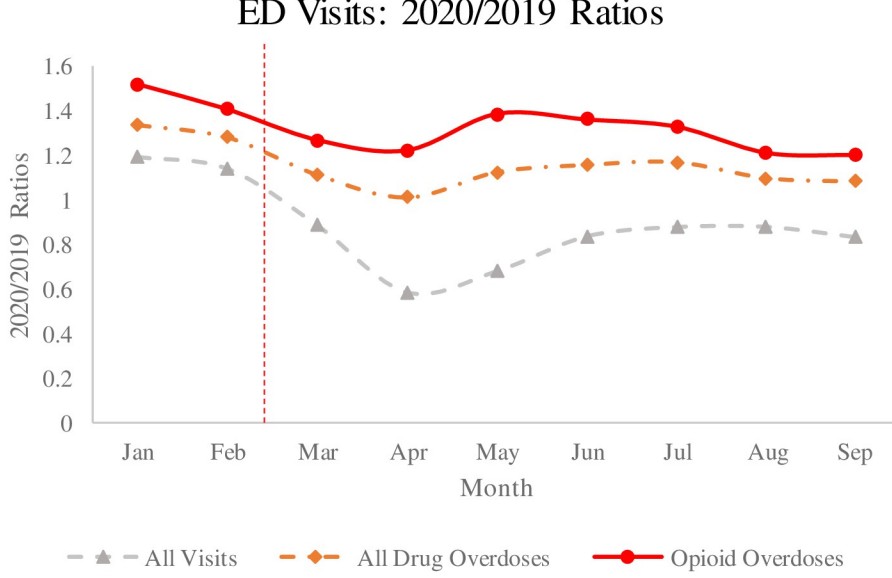

**Panel B**

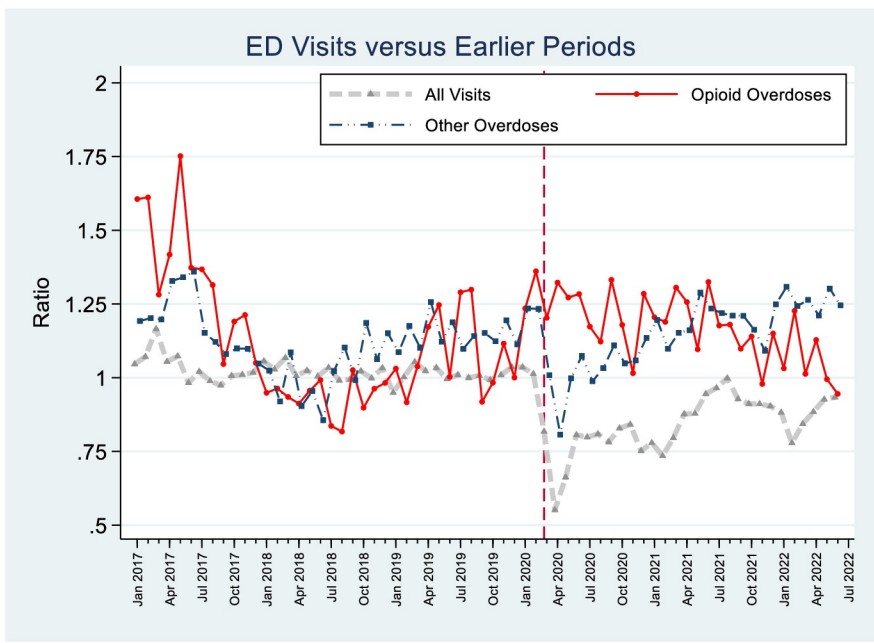

**Fig 3. Monthly ratios of ED visits during the COVID-19 pandemic. Panel A**. *National Data*. shows Monthly Ratios of 2020 to 2019 emergency departments visits for January 2019 through September 2020 for all visits, all drug overdose visits, and opioid overdose visits using CDC data. **Panel B**. Ratios for all visits, opioid overdose visits, and all other overdose visits for January 2017 through October 2021 using data from 181 EDs in 24 states staffed by a national emergency department staffing company. Ratios for 2020–2021 are to same month in 2019. **Panel A. National ED visits for non-fatal overdoses**. **Panel B: Visits to EDs Staffed by National ED Staffing Company**.

whole similar in the pre-pandemic and pandemic periods signal a steadily worsening overdose epidemic. However, they do not send a strong signal that the pandemic affected overdose rates.

Fig 3 shows drug-related ED visit rate ratios using our two data sources. Panel A presents national data for January-September 2020. It shows monthly ratios of 2020 visits to the same month in 2019 for all ED visits, all drug-overdose visits, and opioid-related visits. Year-over-year ratios for all overdose visits and opioid-related visits were well above 1.00 in early 2020, dropped early in the pandemic period, and then partly rebounded, but remained below the January-February ratios. The smaller early-pandemic decline for drug-overdose visits than for all visits is consistent with overdose visits being relatively acute and not easily avoidable.

Panel B presents ED data from the national physician staffing group. It shows monthly ratios (current month versus same month in the previous year (except that we compared 2021 and 2022 to 2019). The figure shows data for all visits regardless of cause (for comparison purposes), data for opioid-related overdose visits and all other drug overdose visits over 2017-June 2022. S1 Appendix, Fig App-2 includes 95% CIs.

For opioid-related visits, the year-over-year ratios were well above 1.00 in 2017, fell over 2017 and 2018 to below 1.00, and then to well above 1.00 in the immediate pre-pandemic months of January-February 2020. Visits for other drug overdoses followed a broadly similar pattern. These large fluctuations underscore difficulties in assessing whether a pandemic effect exists.

During the pandemic and immediate pre-pandemic period, opioid-related visit ratios were well above 1.00 throughout, but not notably higher during the pandemic period. Non-opioid drug overdose visit ratios fell sharply in March-April 2020, followed by a rebound that reached pre-pandemic ratios by mid 2021.

Thus, in both the national and data and the longer tome series available from the national physician staffing group, there is evidence of a severe and worsening opioid epidemic, but the time patterns do no suggest that the COVID-19 pandemic had a major effect on the course of the opioid epidemic.

## Discussion

Prior studies have linked higher overdose mortality and overdose-related ED visits in 2020 primarily to the COVID-19 pandemic [2–11]. This led to a narrative in public health, academic, and media circles that attributes much of the rise in overdose mortality in 2020 and 2021 to the pandemic. If accepted, that narrative deflects public attention from a direct response to overdose deaths, particularly synthetic-opioid deaths.

It is conceptually plausible that the social and healthcare disruptions of the COVID-19 pandemic could have led to higher drug use, less substance abuse treatment (leading indirectly to higher use), or both. Attendance at substance abuse centers indeed fell substantially in 2020 [32].

But the role of the pandemic in contributing to overdose deaths and overdose-related ED visits is an empirical question. Prior studies of overdose deaths and overdose-related ED visits do not model the counterfactual of expected overdose deaths and ED visits during the pandemic period, based on pre-pandemic trends. Yet, opioid-related overdose deaths and overdose-related ED visits were rising before the pandemic, especially beginning in the second half of 2019, with the fentanyl supply reaching new areas.

We investigated the extent to which pre-pandemic trends, extrapolated to the pandemic period, can explain higher pandemic-period drug-overdose deaths and ED visits. Given large pre-pandemic variation in overdose rates, both across time and across states, we limited the

extrapolation to the first pandemic year. In our judgment, one cannot credibly predict the counterfactual for a longer period, given the many time-varying factors that could affect overdose mortality.

During the first pandemic year, the extrapolation models predicted overdose levels similar to observed levels. Our evidence taken as a whole, including the evidence on placebo pandemic-onset months, and the state-level evidence showing large variation in actual-minus-predicted overdose mortality, support a continuation of pre-pandemic trends as a principal driver of rising mortality. We cannot rule out the pandemic as a contributing factor, but the magnitude of a pandemic effect seems likely to be limited. This conclusion applies if one posits an immediate onset of pandemic-related effects in or soon after March 2020. It would apply even more strongly if one posits a delayed effect of the pandemic, because it would be still harder for a hypothesized pandemic effect to catch up to the pandemic-period rise that would be expected based on pre-pandemic trends.

Several pieces of evidence support this assessment. First, the rise in mortality in 2020 and 2021 comes entirely from synthetic opioids. Yet, if the pandemic and lockdowns were an important cause of drug use, we might expect mortality to rise for other drugs as well.

Second, we found evidence of generally rising ED visits for drug overdoses during both the pandemic and the immediate pre-pandemic period, relative to the same month a year earlier, but no change in ED visit ratios versus the prior year, around the pandemic onset.

Our results differ from prior studies partly because our data are not limited to a single geographic area and cover a longer time period. Importantly, our results differ because we carefully model the counterfactual of what level of overdose deaths and ED visits would have been expected without the pandemic. A simple comparison of pandemic-period rates to pre-pandemic rates, without estimating the counterfactual, can be misleading. For example, we find that Illinois overdose deaths during the pandemic were *below* those predicted by our model. This is contrary to the core result from the Mason et al. (2020) study of Cook County, Illinois, which simply compared pandemic-period to pre-pandemic averages [3]. We also find that Ohio overdose deaths in the first pandemic year were lower than predicted, even though Currie et al. (2021) find a short-lived increase in synthetic-opioid deaths, early in the pandemic. No prior study adequately uses pre-pandemic data to model the extent to which pandemic-period outcomes can be predicted by pre-pandemic trends in overdose deaths. Only one prior study uses pre-pandemic data to model counterfactual outcomes during the pandemic [4], and this study uses an unrealistic model which predicts declining overdose deaths in 2020 despite rising deaths in the second half of 2019. Most studies focus on limited geographic areas and/or short time periods [2, 3, 5, 7, 10–12, 25].

For ED visits, prior studies did not account for rising ED visit rates in the pre-pandemic period. If one compares visit rates to the same period a year earlier, visit rate ratios are generally above one, in both the pre-pandemic and pandemic periods. But we find no evidence that this ratios rises in the pandemic period. This finding of rising visit rates, but no change in trend, is consistent with the two prior multistate studies [8, 9]. The other studies cover limited geographic areas, short time periods, or both [10–12].

Our models of counterfactual deaths, based on pre-pandemic trends, predict 24.7–26.3 drug-related deaths per 100,000 population in 2020, versus the actual 25.5. For the first three months of 2021, the 2017 Model predicts actual mortality fairly well, while the 2015 Model predicts higher mortality than was observed. Our analysis of placebo pandemic-onset months shows no evidence of a change in trend when the pandemic began in March 2020.

At the state-level, we find substantial heterogeneity in actual-minus-predicted mortality. This suggests that state-specific characteristics are important predictors of mortality trends. In some states actual opioid deaths during the pandemic exceed predicted deaths, but in a

roughly equal number of states, actual deaths during the pandemic were less than predicted, using a model fitted to state-level pre-pandemic data.

Our results provide evidence that much and perhaps all of the increase in drug-related overdose deaths during the pandemic period can be explained by continuation of pre-pandemic trends in opioid mortality. If overdose mortality during the pandemic in fact largely reflects pre-pandemic trends, then policymakers, health care providers, and the public should not expect drug overdose deaths to decline as the pandemic recedes. Multi-modal strategies are needed to address the opioid epidemic, including addressing fentanyl supply and mixing heroin and other drugs with fentanyl. This includes: increased surveillance of illicit markets to monitor spread of fentanyl; deploying pill-testing technology, which can allow users to test for fentanyl, which could give suppliers incentives not to add fentanyl to their products [33]; and naloxone distribution programs to increase the likelihood of survival after an overdose [34].

Our claim is not that the pandemic had no effect on overdose mortality, but instead that (i) the magnitude and existence of any effect is unknown and will likely remain so; (ii) we provide suggestive evidence that the pandemic effect could have been much smaller than suggested in other studies which do not carefully model the counterfactual; and (iii) results using data from one or a few states are suspect given the variation across states in how opioid mortality changed during the pandemic period. We do not believe that the data will support a more definitive message than that.

## Limitations

Our model is not causal. We can neither rule out nor rule in a causal effect of fentanyl access or the pandemic on overdose deaths. A causal design requires a control group. However, the pandemic onset was basically national, at the same point in time (March 2020), with a wave of lockdowns and restrictions that were adopted in most states at similar times, and relaxed at similar times. There are some variations in the timing and extent of state-level restrictions, but we concluded that the variation was too limited, relative to the state-level heterogeneity, to make feasible a classic difference-in-differences design, relying on state-level variation in regulation extent or timing. The difficulty in estimating counterfactual trends is compounded by the CDC practice of reporting only rolling 12-month mortality, instead of monthly totals; this makes it harder to identify trend changes in the data.

We conclude, more mildly, that: (i) our evidence is consistent with a causal effect fentanyl access; (ii) the evidence for a pandemic effect is weak; and (iii) a strong case for a pandemic effect likely cannot be made, given the challenges in estimating the counterfactual of overdose mortality without the pandemic.

Estimating the counterfactual is difficult. The difficulty is compounded because the CDC releases only rolling 12-month totals, instead of monthly mortality rates. Our model of counterfactual trends can provide only rough estimates of counterfactual mortality, increasingly so as the extrapolation period gets longer. We therefore model only the first pandemic year, through March 2021.

We rely on data for a limited pre-pandemic period but given the rapid evolution of the opioid epidemic, it is unclear whether a longer pre-period would meaningfully improve predictive accuracy. The fourth-order polynomial we used does a good job of capturing pre-pandemic trends over 2015-early 2020, but overfitting, leading to poor prediction during the pandemic period, remains an unavoidable concern. Using a higher-order polynomial to fit pre-pandemic data would increase overfitting risk, and would not change our main results.

Overdose mortality may be measured with error, and the preliminary counts we rely on may differ from final counts but we see no reason why these factors should strongly affect mortality trends.

Our measure of overdose-related ED visits cannot capture all of the reasons why the pandemic may have affected ED visit rates. Our data does not capture the ED visits that did not happen because of ED avoidance, but ED-avoidance was likely limited primarily to the period immediately following pandemic onset [11].

Our datasets have limitations. CDC national and state mortality data is available only as a rolling 12-month total, and only as exact counts, without confidence intervals. The national ED-visit data addresses a limited time period, through September 2020. The ED-staffing-company data is limited to 181 EDs in 24 states and may not be nationally representative.

## Conclusion

The opioid epidemic is a major and growing crisis. It's effect on life expectancy far exceeds the effect of the COVID-19 pandemic and remains with us as the pandemic recedes [35, 36]. Yet during the last several years, the opioid epidemic has been overshadowed by the COVID-19 pandemic, and may not be receiving the attention it deserves from either the public at large or from public health departments.

Prior work treats the pandemic as a major driver of rising overdose deaths, but has important data and methodology limitations. In contrast, we provide evidence that the effect of the COVID-19 pandemic on the opioid epidemic is uncertain, and could be small. The major drivers of rising overdose deaths likely lie elsewhere.

## Supporting information

**S1 Appendix.**
(DOCX)

**S1 File. Drug mortality database.**
(TXT)

**S2 File. National suspected non-fatal overdose ED visits.**
(CSV)

**S3 File. Drug mortality analysis do file.**
(TXT)

## Author Contributions

**Conceptualization:** Ali Moghtaderi, Jesse M. Pines, Arvind Venkat, Bernard Black.

**Data curation:** Ali Moghtaderi, Mark S. Zocchi, Bernard Black.

**Formal analysis:** Ali Moghtaderi, Mark S. Zocchi, Jesse M. Pines, Bernard Black.

**Investigation:** Ali Moghtaderi.

**Methodology:** Ali Moghtaderi, Mark S. Zocchi, Jesse M. Pines, Arvind Venkat, Bernard Black.

**Project administration:** Ali Moghtaderi.

**Software:** Ali Moghtaderi.

**Supervision:** Ali Moghtaderi.

**Visualization:** Ali Moghtaderi, Bernard Black.

**Writing – original draft:** Ali Moghtaderi.

**Writing – review & editing:** Ali Moghtaderi, Mark S. Zocchi, Jesse M. Pines, Arvind Venkat, Bernard Black.

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
