## [Decision Letter · Decision Letter 0]

12 Mar 2023

PONE-D-23-01497Estimating The Uncertain Effect of the COVID Pandemic on Drug OverdosesPLOS ONE

Dear Dr. Moghtaderi,

Thank you for submitting your manuscript to PLOS ONE. After careful consideration, we feel that it has merit but does not fully meet PLOS ONE’s publication criteria as it currently stands. Therefore, we invite you to submit a revised version of the manuscript that addresses the points raised during the review process.

 Please address the comments of reviewer 2, with special attention to ecological fallacy and the composition of the sample.

We look forward to receiving your revised manuscript.

Kind regards,

Vincenzo Alfano

Academic Editor

PLOS ONE

Journal Requirements:

Reviewers' comments:

Reviewer's Responses to Questions

**Comments to the Author**

1. Is the manuscript technically sound, and do the data support the conclusions?

Reviewer #1: Partly

Reviewer #2: Yes

2. Has the statistical analysis been performed appropriately and rigorously? 

Reviewer #1: Yes

Reviewer #2: Yes

3. Have the authors made all data underlying the findings in their manuscript fully available?

Reviewer #1: No

Reviewer #2: Yes

4. Is the manuscript presented in an intelligible fashion and written in standard English?

Reviewer #1: No

Reviewer #2: Yes

5. Review Comments to the Author

Reviewer #1: This manuscript examined if increases in overdose deaths and ED visits were associated to the COVID-19 pandemic, increased synthetic-opioid use or both. The study takes into account pre-pandemic estimations to determine the effect of COVID-19 on Opioid overdose deaths and ED visits, which is novel and accurate comparing previous studies. Nonetheless, the manuscript needs clarity, as at times redundant and is not concise, which dilutes the findings.

For example, there paragraphs: “Figure 1, Panel A, presents 12-month rolling average rates for all drug-related overdose deaths, synthetic-opioid-related deaths, and other overdose deaths, per 100,000 population”, but this type of information pertains to the figure, and results should be concise and state the statistical significance of what is added in the tables/figures, which is lacking in this section.

Additionally, the authors tend to be speculative at times in their discussion, for example using words like (i.e. somewhat lower), but don't back up these statements (is it statistically significant?)

Finally, there are moments where the authors state causality, but acknowledge that there could be other potential factors (such as the pandemic) that could be contributing to the predicted trends. With that said, the authors should avoid any causal jargon (even though the statistical model considers counterfactuals) and focus their study on the predictions found.

Reviewer #2: The authors provide an in-depth and thorough review of the literature in their introduction that is relevant to their research question, and they provide adequate rationale for their study at hand.

What was the definition of death or non-death opioid presentations. Was this data determined from diagnosis codes used for ED visits, was this a clnical diagnosis, post-mortem, utilizing confirmatory blood levels? May be worth specifying.

Adequate explanation of limitations provided throughout, which is appreciated.

I believe the author's conclusions to be sound, and not overstated. There are limitations to the data presented, as the author's suggest; however, they do not drive their conclusions too far.

6. PLOS authors have the option to publish the peer review history of their article (what does this mean?). If published, this will include your full peer review and any attached files.

Reviewer #1: No

Reviewer #2: No

---

## [Author Response · Author response to Decision Letter 0]

27 Apr 2023

Dear Professor Alfano,

Thank you for consideration of our paper, “Estimating The Uncertain Effect of the COVID Pandemic on Drug Overdoses.” We would like to thank the reviewers for their thoughtful comments. We believe our paper has substantially improved because of their comments. Below, we outline our responses to the comments. 

Please address the comments of reviewer 2, with special attention to ecological fallacy and the composition of the sample.

Reply. We respond to the referees below.

Ecological fallacy. Neither referee mentions an ecological fallacy issue, and we do not believe our study has this problem. This term is usually applies to situations where one infers individual-level characteristics from group results. We do not do this.

Sample composition. We rely partly on CDC data and partly on data from US Acute Care Solutions. We reviewed what we said about each data source, and revised for clarity. For overdose deaths, the national data is complete. For national emergency department (ED) visit data, the CDC draws on EDs from 42 states that submit data to it. The sample should be reasonably representative of larger EDs, but we don’t want to claim that because the CDC does not disclose which EDs participate. For both of our sources for ED visit data (CDC and the ED staffing company), we added information about visit counts to the text.

Reply. We now provide the data on drug overdose mortality and the national CDC data on non-fatal drug ED visits, as well as our codes in our resubmission package. We are not allowed to share the data from the national emergency staffing group based on the data use agreement in place.

Reply. Done.

Reviewer's Responses to Questions

Comments to the Author

1. Is the manuscript technically sound, and do the data support the conclusions?

Reviewer #1: Partly

Reviewer #2: Yes

2. Has the statistical analysis been performed appropriately and rigorously? 

Reviewer #1: Yes

Reviewer #2: Yes

3. Have the authors made all data underlying the findings in their manuscript fully available?

Reviewer #1: No

Reviewer #2: Yes

4. Is the manuscript presented in an intelligible fashion and written in standard English?

Reviewer #1: No

Reviewer #2: Yes

5. Review Comments to the Author

Reviewer #1: 

This manuscript examined if increases in overdose deaths and ED visits were associated to the COVID-19 pandemic, increased synthetic-opioid use or both. The study takes into account pre-pandemic estimations to determine the effect of COVID-19 on Opioid overdose deaths and ED visits, which is novel and accurate comparing previous studies. Nonetheless, the manuscript needs clarity, as at times redundant and is not concise, which dilutes the findings.

Reply. We reread the manuscript, and tried to eliminate redundancy.

For example, there paragraphs: “Figure 1, Panel A, presents 12-month rolling average rates for all drug-related overdose deaths, synthetic-opioid-related deaths, and other overdose deaths, per 100,000 population”, but this type of information pertains to the figure, and results should be concise and state the statistical significance of what is added in the tables/figures, which is lacking in this section.

Reply. We added additional explanation to the text of the implications of this figure. This figure reports national totals as reported by the CDC; there are no associated confidence intervals. The CDC does not report them and we cannot compute them from the data that the CDC provides. We added an explanation to the limitations section that confidence intervals around the CDC-reported data are not available. 

Additionally, the authors tend to be speculative at times in their discussion, for example using words like (i.e. somewhat lower), but don't back up these statements (is it statistically significant?)

Reply. We compare predicted deaths from our models (based on pre-pandemic trends) at different points in time, to actual mortality data from the CDC. The CDC data does not provide confidence intervals, so our models do not have them either. Thus, terms like “somewhat lower” are the best we can do, in describing how the prediction compares to the actual data. 

We tried to be clear that any comparison of actual to predicted mortality is speculative.

Finally, there are moments where the authors state causality, but acknowledge that there could be other potential factors (such as the pandemic) that could be contributing to the predicted trends. With that said, the authors should avoid any causal jargon (even though the statistical model considers counterfactuals) and focus their study on the predictions found.

Reply. We tried to avoid any causal claims. For example, the draft stated that “Our model is not causal. Thus, we can neither rule out nor rule in a causal effect of fentanyl access or the pandemic on overdose deaths.” We added a similar statement to the introduction, saying we do not have a control group and therefore cannot have a causal research design. We reviewed the draft and the words used and revised for clarity. We believe that readers will understand that we are not making causal claims.

Reviewer #2:

The authors provide an in-depth and thorough review of the literature in their introduction that is relevant to their research question, and they provide adequate rationale for their study at hand.

What was the definition of death or non-death opioid presentations. Was this data determined from diagnosis codes used for ED visits, was this a clinical diagnosis, post-mortem, utilizing confirmatory blood levels? May be worth specifying.

Reply. Deaths come from the CDC, which relies on the text fields included in death certificates. The National Center for Health Statistics, within the CDC, received these death certificates, and assigns ICD-10 codes for primary cause of death and secondary causes based on the text fields. How accurate the death certificates are, and what blood tests or other laboratory results they are based on, is not known, and may vary based on who fills out the death certificate and customary practice where the person died. We explain this in the revised draft.

For ED visits, counts are based on diagnosis codes assigned by the clinician (usually an ED physician) at the time of the visit. We revised the draft to add details on this dataset.

Adequate explanation of limitations provided throughout, which is appreciated.

Reply. We added a limitation with regard to the accuracy of death certificates.

I believe the author's conclusions to be sound, and not overstated. There are limitations to the data presented, as the author's suggest; however, they do not drive their conclusions too far.

---

## [Decision Letter · Decision Letter 1]

22 May 2023

PONE-D-23-01497R1Estimating The Uncertain Effect of the COVID Pandemic on Drug OverdosesPLOS ONE

Dear Dr. Moghtaderi,

Thank you for submitting your manuscript to PLOS ONE. After careful consideration, we feel that it has merit but does not fully meet PLOS ONE’s publication criteria as it currently stands. Therefore, we invite you to submit a revised version of the manuscript that addresses the points raised during the review process.

We look forward to receiving your revised manuscript.

Kind regards,

Vincenzo Alfano

Academic Editor

PLOS ONE

Reviewers' comments:

Reviewer's Responses to Questions

**Comments to the Author**

1. If the authors have adequately addressed your comments raised in a previous round of review and you feel that this manuscript is now acceptable for publication, you may indicate that here to bypass the “Comments to the Author” section, enter your conflict of interest statement in the “Confidential to Editor” section, and submit your "Accept" recommendation.

Reviewer #1: (No Response)

2. Is the manuscript technically sound, and do the data support the conclusions?

Reviewer #1: Yes

3. Has the statistical analysis been performed appropriately and rigorously? 

Reviewer #1: Yes

4. Have the authors made all data underlying the findings in their manuscript fully available?

Reviewer #1: Yes

5. Is the manuscript presented in an intelligible fashion and written in standard English?

Reviewer #1: No

6. Review Comments to the Author

Reviewer #1: The authors addressed most of the requests from the first review, which is appreciated. But the manuscript is still too long and needs to be concise and clear.

For example, the first five and a half pages are introduction (even though the authors tried to use different headings). This information can be summarized in half of the length and some of this information (for example evidence on overdose mortality) can be use in the discussion, which in comparison is short and should be the focus of the paper along the results (half the length).

Same with the methodology. The authors describe the different analyses performed in pages 9-10, but then go into detail in the statistical analysis. This is repetitive.

Results are showing the same issue, and it was not accounted from the previous review. For example: "Fig 1, Panel A, presents 12-month rolling average rates for all drug-related overdose deaths, synthetic-opioid-related deaths, and other overdose deaths, per 100,000 population, from 2015-2021". should be considered a footnote in the figure and not be in the main results. This just distract the reader from the main findings (which would be: ...rise during the pandemic period continues a trend toward rising synthetic opioid deaths that accelerated in mid-2019. In contrast, other overdose deaths were basically flat from mid-2019 on).

7. PLOS authors have the option to publish the peer review history of their article (what does this mean?). If published, this will include your full peer review and any attached files.

Reviewer #1: No

---

## [Author Response · Author response to Decision Letter 1]

29 Jun 2023

Review Comments to the Author

Reviewer #1: The authors addressed most of the requests from the first review, which is appreciated. But the manuscript is still too long and needs to be concise and clear.

For example, the first five and a half pages are introduction (even though the authors tried to use different headings). This information can be summarized in half of the length and some of this information (for example evidence on overdose mortality) can be use in the discussion, which in comparison is short and should be the focus of the paper along the results (half the length).

Reply. We moved the detailed description of the prior literature to the Appendix; the introduction is now substantially shorter. 

Same with the methodology. The authors describe the different analyses performed in pages 9-10, but then go into detail in the statistical analysis. This is repetitive.

Reply. We combined these two sections. The new “Methods” section describes the databases we use first, then provides information about the outcomes we study, and finally describes the statistical analysis. 

Results are showing the same issue, and it was not accounted from the previous review. For example: "Fig 1, Panel A, presents 12-month rolling average rates for all drug-related overdose deaths, synthetic-opioid-related deaths, and other overdose deaths, per 100,000 population, from 2015-2021". should be considered a footnote in the figure and not be in the main results. This just distract the reader from the main findings (which would be: ...rise during the pandemic period continues a trend toward rising synthetic opioid deaths that accelerated in mid-2019. In contrast, other overdose deaths were basically flat from mid-2019 on).

Reply. We revised and shortened this section.

---

## [Decision Letter · Decision Letter 2]

17 Jul 2023

Estimating The Uncertain Effect of the COVID Pandemic on Drug Overdoses

PONE-D-23-01497R2

Dear Dr. Moghtaderi,

We’re pleased to inform you that your manuscript has been judged scientifically suitable for publication and will be formally accepted for publication once it meets all outstanding technical requirements.

Kind regards,

Vincenzo Alfano

Academic Editor

PLOS ONE

Additional Editor Comments (optional):

Reviewers' comments:

Reviewer's Responses to Questions

**Comments to the Author**

1. If the authors have adequately addressed your comments raised in a previous round of review and you feel that this manuscript is now acceptable for publication, you may indicate that here to bypass the “Comments to the Author” section, enter your conflict of interest statement in the “Confidential to Editor” section, and submit your "Accept" recommendation.

Reviewer #1: All comments have been addressed

2. Is the manuscript technically sound, and do the data support the conclusions?

Reviewer #1: Yes

3. Has the statistical analysis been performed appropriately and rigorously? 

Reviewer #1: Yes

4. Have the authors made all data underlying the findings in their manuscript fully available?

Reviewer #1: Yes

5. Is the manuscript presented in an intelligible fashion and written in standard English?

Reviewer #1: Yes

6. Review Comments to the Author

Reviewer #1: (No Response)

7. PLOS authors have the option to publish the peer review history of their article (what does this mean?). If published, this will include your full peer review and any attached files.

Reviewer #1: No

---

## [Editor Report · Acceptance letter]

31 Jul 2023

PONE-D-23-01497R2 

Estimating The Uncertain Effect of the COVID Pandemic on Drug Overdoses 

Dear Dr. Moghtaderi:

I'm pleased to inform you that your manuscript has been deemed suitable for publication in PLOS ONE. Congratulations! Your manuscript is now with our production department. 

Kind regards, 

on behalf of

Dr. Vincenzo Alfano 

Academic Editor

PLOS ONE